# Physical Activity on Prescription in Routine Health Care: 1-Year Follow-Up of Patients with and without Counsellor Support

**DOI:** 10.3390/ijerph17165679

**Published:** 2020-08-06

**Authors:** Pia Andersen, Sara Holmberg, Kristofer Årestedt, Lena Lendahls, Per Nilsen

**Affiliations:** 1Department of Research and Development, Region Kronoberg, SE-351 88 Växjö, Sweden; sara.holmberg@kronoberg.se (S.H.); lena.lendahls@lnu.se (L.L.); 2Department of Health, Medicine and Caring Sciences, Division of Society and Health, Linköping University, SE-581 83 Linköping, Sweden; per.nilsen@liu.se; 3Division of Occupational and Environmental Medicine, Institute of Laboratory Medicine, Lund University, SE-221 00 Lund, Sweden; 4Faculty of Health and Life Sciences, Department of Medicine and Optometry, Linnaeus University, SE-391 82 Kalmar, Sweden; 5Faculty of Health and Life Sciences, Department of Health and Caring Sciences, Linnaeus University, SE-391 82 Kalmar, Sweden; kristofer.arestedt@lnu.se; 6The Research Section, Region Kalmar County, SE-391 26 Kalmar, Sweden

**Keywords:** physical activity on prescription, counsellor, counselling, health care, patients

## Abstract

The effectiveness of counsellor support in addition to physical activity on prescription (PAP) from health care professionals has rarely been evaluated. This observational follow-up study investigated differences in physical activity levels and health-related quality of life (HRQoL) one year after PAP regarding patients’ use of counsellor support in addition to PAP in routine care. The study was conducted in a Swedish health care region in which all patients receiving PAP from health care professionals were offered counsellor support. Data were collected from medical records and questionnaires (baseline and follow-up). Of the 400 study participants, 37% used counsellor support. The group of counsellor users attained a higher level of physical activity one year after receiving PAP compared to the group of non-users (*p* < 0.001). The level of physical activity was measured by a validated index (score 3–19) calculated from weekly everyday activity and exercise training. Comparison of the change in scores between baseline and follow-up showed a significant difference between the two groups, (*p* < 0.001). The median difference in the PAP + C group was 2.0 (interquartile range, 7.0) and 0.0. among non-users (interquartile range, 4.0). Significant differences in HRQoL were due to positive improvements among counsellor users, with the main improvement in general health. The conclusion is that patients using counsellor support after receiving PAP from health care professionals had higher physical activity and better HRQoL one year after compared with patients who did not use this support.

## 1. Introduction

Physical activity on prescription (PAP) interventions including health care professionals’ prescription of physical activity or a physical activity referral to an exercise provider outside the health care system have been developed for health care services to support patients who would benefit from increased physical activity [1,2]. Several studies on PAP interventions have reported positive outcomes on patients’ physical activity levels and health-related quality of life (HRQoL) [3,4,5,6]. Longer duration of PAP interventions is usually associated with a higher level of physical activity [7]. However, poor patient adherence is typically a problem with long-term maintenance of behavior change interventions [8]. Research has documented that poor long-term adherence is a common problem for PAP interventions [7,9,10].

Recognizing the challenges of long-term adherence, some PAP interventions such as the Swedish PAP [11] have been developed based on individually tailored written prescriptions of activities delivered with a patient-centered approach, i.e., delivery of care that is respectful, individualized and empowering, to encourage individual participation of the patient and a patient–caregiver relationship of mutual trust, empathy and shared knowledge [12]. Such interventions show promise with regard to achieving positive effects on patients’ physical activity levels [13,14]. More tailored approaches to PAP delivery are facilitated by multi-professional organizations [15]. In such organizations, patients are allowed to receive counselling support by an exercise specialist within a sport organization [13] or from an allied health care professional in routine health care, e.g., a physiotherapist [5,16].

Despite promising findings regarding more tailored and patient-centered approaches to the delivery of PAP, there is a paucity of research investigating the outcomes on patients’ physical activity levels regarding counselling support in addition to PAP. This study addresses this knowledge gap in the PAP literature by investigating the impact on patients’ physical activity levels and HRQoL of individualized counselling support provided by health care professionals in addition to PAP. The aim of this study was to investigate differences in physical activity levels and HRQoL one year after PAP among patients who used physical counsellor support in addition to PAP and patients who did not accept this offer of support and only received PAP.

## 2. Materials and Methods

### 2.1. Study Design

This is an observational follow-up study of a routine health care PAP intervention including an offer of physical activity counsellor support for all PAP recipients in addition to health care professional’s prescription of physical activity. The study was developed in accordance with the ethical principles of the Declaration of Helsinki and was approved by the Regional Ethical Review Board in Linköping (No. 2013/51-31).

### 2.2. Setting

The study was conducted in Region Kronoberg, a health care region located in southern Sweden. This health care region includes 22 public and 11 privately operated primary health care centers (all publicly funded), two public hospitals for specialized somatic care and one for specialized psychiatric care. The health care region delivers care to the whole population in Kronoberg County, approximately 200,000 people. The PAP intervention has been in use this health care organization since 2009 and is an adapted model of the Swedish PAP program.

The Swedish PAP program includes patient-centered counselling, written prescription of individualized physical activity using the Physical Activity in the Prevention and Treatment of Disease handbook [17], collaboration with local sport organizations and follow-up assessments of patients’ physical activity levels [11]. There are no recommendations on best practice models for delivery of support after prescription of physical activity in the Swedish PAP program, and different modes of support delivery exist. Support can for instance be delivered within health care or by sports organizations, to all PAP recipients, or as an offer for those who believe they need additional support.

The PAP intervention investigated in this study includes a multi-professional organization for routine health care delivery of PAP involving health care professionals who prescribe physical activity and counsellors who delivers support after PAP. The rationale for implementation of physical activity counsellor support was to provide simplified routines for health care professionals’ delivery of PAP and to offer high-quality physical activity counselling, with the aim of reaching a larger population who would benefit from physical activity and increasing the likelihood of successful results. The physical activity counsellors are employed and financed by the Region Kronoberg for the specific task of supporting patients who are prescribed PAP. They are licensed health care professionals (e.g., nurses or physiotherapists) and are trained in motivational interviewing techniques [18]. Motivational interviewing training available for the physical activity counsellors varies from a course lasting for several weeks at the university to a four-day short course administered by the Region Kronoberg. One-day workshops are offered every other year. Training in motivational interviewing is obligatory for physical activity counsellors but not for the health care professionals who prescribes PAP.

Patients who would benefit from increased physical activity are identified and prescribed PAP during routine health care visits and during in-patient hospital stays by licensed health care professionals, e.g., physicians, nurses, and physiotherapists. When health care professionals prescribe PAP, they are expected to have a short dialogue about physical activity with the patient (5–10 min) based on the individual’s need of physical activity, prescribe a written electronic PAP, including the reason for the prescription and any contraindications for activity. There is no obligation on the health care professional who prescribes the PAP to specify type and dose of physical activity, because this can also be managed by the physical activity counsellors. Patients receive a copy of the written PAP along with written information about how to get in contact with a physical activity counsellor.

The next step for patients who want counsellor support is to phone or e-mail a counsellor to schedule a face-to-face counselling session. Time scheduled for the counselling is between 30 and 60 min, although the first visit is always 60 min. There are no patient fees for counsellor support and there is no predetermined number of counselling visits. The number of follow-up counselling sessions is based on the individual PAP recipient’s need of support. In the years of 2009 to 2012 one-third of all PAP recipients used counsellor support [19], and the min-max number of visits within these years were 1 to 12 (mean = 3.5 [SD = 1.8]).

Counsellor support is to be utilized within one year after prescription. If appropriate, the physical activity counsellor encourages a wider health dialogue, e.g., dietary habits, tobacco use, and alcohol consumption.

PAP recipients can be recommended everyday activities or exercise training. The activities can be performed on one’s own or in local sport clubs. There are no differences in access to activities for patients who use or do not use counsellor support. Fees for participating in local sport clubs are paid by the patient. Some sport clubs have reduced fees for a period.

### 2.3. Recruitment Process

All patients receiving PAP from June 2013 until June 2014 were identified through the electronic medical record system and recruited consecutively by postal invitations. Inclusion criteria were ≥18 years of age and prescribed PAP in primary care or specialized care (somatic or psychiatric hospital care). A few patients were excluded due to prescription from other settings such as occupational health services and elderly care. Two primary health care centers did not agree to be involved in the study and, therefore, patients receiving PAP from these centers were not invited. A postal invitation was sent to the patient 2 to 3 weeks after the PAP prescription date. One reminder was sent to non-responders after 2 to 3 weeks. A total 1503 patients were invited to participate (Figure 1). The invitation included information about the study, a baseline questionnaire, and a pre-paid return envelope. A total of 604 patients responded at baseline (40%). A follow-up questionnaire and a pre-paid return envelope were sent after one year to those who responded at baseline. One reminder was sent to non-responders after 2 to 3 weeks. Of the 604 patients who participated at baseline, 400 responded at follow-up (66%) (Figure 1). Among these responders, 149 patients had used counsellor support (the PAP + C group) and 251 patients had not used this support (the PAP-only group).

### 2.4. Data Collection

Data were collected from the electronic medical record system and from the baseline and the follow-up questionnaire. The same electronic medical record system was used by all health care settings. The following patient characteristics were obtained from the medical record data: sex, age, registered diagnoses according to the International Classification of Diseases version 10 (ICD-10) [20], out-patient health care visits and in-patient hospital care, prescribing setting, prescribing profession, and the number of visits to physical activity counsellors. Data on diagnoses and health care consumption were collected for the 12 months before the PAP prescription date. The rationale for the categorization of diagnostic groups, i.e., the ICD-10 classified disease groups, and the categorization of health care consumption was to correspond to higher or lower morbidity rates, including multi-morbidity. To capture all visits to a physical activity counsellor data were obtained for the 16 months after the PAP prescription date. The questionnaire included data on socio-demographics, self-rated physical activity, physical activity habit strength, confidence to change activity, perceptions of importance to change activity and HRQoL.

### 2.5. Primary and Secondary Outcomes

The primary outcome was the level of physical activity during a regular week, as measured using the Swedish National Board of Health and Welfare questions recommended for clinical practice: “During a regular week, how much time are you physically active in ways that are not exercise, for example walks, bicycling, or gardening? Add together all activities lasting at least 10 min”; and “During a regular week, how much time do you spend exercising on a level that makes you short winded, for example, running, fitness class, or ball games?” [21]. The response format of these questions is categorical, and the alternatives includes different levels of activity in minutes. The everyday activity item has seven numbered response alternatives (“1” implies no activity while “7” implies more than 300 activity minutes during a regular week) and the exercise training item has six numbered alternatives (“1” implies no exercise while “6” implies more than 120 exercise minutes during a regular week) [21,22]. The total level of weekly physical activity is calculated from the response of these two questions: everyday activity + (exercise training × 2), which gives a possible score range of 3 to 19 [21,22]. The highest score 19 is equal to >300 min of everyday activity and >120 min of exercise training per week, and the lowest score 3 is equal to no activity minutes [22]. A score of 11 corresponds to at least 150 min of moderate intensity physical activity throughout a week measured by accelerometer [21,22]. This scale has been validated for identifying insufficient activity (sensitivity) and sufficient activity (specificity) [21]. The categorical answer modes of the physical activity questions correlate with accelerometer data, and validity has been found to be in line with several other extensively used self-reported physical activity measures [22]. 

Secondary outcomes were physical activity habit strength and HRQoL. Physical activity habit strength was measured using the validated Self-Report Behavioral Automaticity Index (SRBAI) [23]. The participants were asked “Physical activity in leisure time is something... a) I do automatically, b) I do without having to consciously remember, c) I do without thinking, d) I start doing before I realize I’m doing it.” Furthermore, the participants were informed that physical activity could include everyday activities and exercise training activities. Response options for each of the four items were given on a Likert scale, ranging from 1 (strongly disagree) to 7 (strongly agree). The higher the score, the stronger the habit. The SRBAI has demonstrated satisfactory measurement properties [23].

HRQoL was measured using the RAND 36-item Health Survey (RAND-36) [24]. The validated standard scoring algorithm for this instrument was used to calculate the eight health scores: physical functioning; role physical limitation; bodily pain; general health; vitality; social functioning; role emotional limitation; and mental health [25]. Calculation of the subscales in RAND-36 is performed in two steps: (1) all items are scored so that a higher level is equal to better health; (2) all items included in the subscales are averaged together. The health scores range from 0 to 100, where higher scores represent better health state [25].

### 2.6. Statistics

Descriptive statistics were used for presentation of the patients’ characteristics. Dependent on the data, chi-squared statistics, the Mann–Whitney U test or the independent sample t-test was used to compare patient characteristics between the study groups.

The Mann–Whitney U test was used to compare the outcome variables between the two groups at baseline and at follow-up. Difference scores (follow-up—baseline) were also calculated for the outcome variables and compared between the two groups using the Mann–Whitney U test. Cohen’s *r* (*r* = Z/√2) for the independent sample Mann-Whitney U test was used as a measure of effect size (effect size; 0.1, small; 0.3, medium; and 0.5, large) [26]. 

In the analysis of physical activity, only those patients who responded to the questions of everyday activity and exercise training at baseline and at follow-up were included. For the SRBAI, only those responding to at least three of the four items were included. For RAND-36, those who completed at least 50% of the questions within a health domain were included. The scale scores represent the average for all items in the scale that the respondent answered.

A *p*-value < 0.05 was regarded as statistically significant. All statistical analyses were performed with SPSS Statistics for Windows 23.0 (IBM Corp, Armonk, NY, USA).

### 2.7. Dropout Analysis

At baseline, non-responders were significantly younger than participants (mean age = 50.4 vs. 59.6 *p* < 0.001) and they were more frequent prescribed in specialized care (28% vs. 16% *p* = 0.001). No sex differences were detected between the two groups (*p* = 0.300).

Dropouts at follow-up were significantly younger than the study participants (mean age = 54.9 years vs. 62.0 years *p* < 0.001) and used counsellor less frequently (29% vs. 37% *p* = 0.042). Compared with the participants, the group of dropouts at follow-up included significantly more patients with mental health disorders (37% vs. 26% *p* = 0.005) but fewer with circulatory diseases (41% vs. 53% *p* = 0.007). The dropouts did not differ from the study participants with regard to any other variables included in the drop-out analysis (sex, education level, baseline level of everyday activity and exercise training, diagnosis, number of registered diagnoses, number of health care visits and overnight hospital stays, prescribing profession and setting).

## 3. Results

### 3.1. Study Participants

The final sample included 400 participants, 149 in the PAP + C group and 251 in the PAP group. Most of the participants were female (69%), the mean age was 62.0 years, and about one-quarter (27%) had a university education. About three-quarters (73%) of the participants had medical registrations in five or more diagnostic groups; musculoskeletal diagnoses were the most common. More than half (59%) of the participants had 11 or more visits to health care in the year before PAP and about one-quarter had at least one overnight hospital stay. About one-quarter (28%) of the participants received PAP from a physician and most (83%) were within primary health care (Table 1).

Participants in the PAP + C group, were significantly younger (*p* = 0.008), had a higher education level (*p* = 0.019), were less frequently diagnosed with musculoskeletal diseases (*p* = 0.047), and were more frequently prescribed PAP by a physician (*p* = 0.046). At baseline, participants in the PAP + C group perceived that it was more important to change physical activity (*p* ≤ 0.001) (Table 1). The mean number of face-to-face counselling visits for patients using counsellor support was 4.1 visits (SD = 2.1, min-max = 1–11).

### 3.2. Primary Outcomes

Using the index value (score 3–19), based on the questions of everyday activity and exercise training, the PAP + C group had significantly lower levels of total physical activity at baseline compared with the PAP-only group (*p* = 0.001). The opposite finding was shown at follow-up, i.e., the PAP + C group had significantly higher levels of physical activity (*p* < 0.001). Comparison of the difference scores between baseline and follow-up showed a significant difference between the two groups (*p* < 0.001). The median difference between baseline and follow-up in the PAP + C group was 2.0 (interquartile range [IQR], 7.0) and 0.0 in the PAP-only group (IQR = 4.0) (*p* < 0.001). The effect size was small (*r* = 0.259 (Table 2). The proportion in the PAP + C group with increased total physical activity was 62% and the corresponding proportion in the PAP-only group was 38%. Participants in the PAP + C group who reached an activity score of 11 to 19, i.e., within 150 min of moderate to vigorous physical activity at follow-up was 55% and 42% in the PAP-only group.

According to everyday activity, no significant differences between the groups were shown at baseline (*p* = 0.404). At follow-up, significantly higher levels were shown in the PAP + C group (*p* = 0.024). The comparison of the difference scores between baseline and follow-up in everyday activity showed a significant difference between the two groups, with higher improvement in the PAP + C group (*p* = 0.014). The effect size was small (*r* = 0.126) (Table 3). The proportion with increased everyday activity at follow-up was 38% in the PAP + C group and 22% in the PAP-only group.

According to exercise training, there was a significant difference between the groups at baseline, with the highest level seen in the PAP-only group (*p* < 0.001). In contrast, the PAP + C group had significantly higher levels of exercise training at follow-up (*p* < 0.001). The comparison of the difference scores between baseline and follow-up in exercise training showed a significant difference between the two groups, with the highest improvement in the PAP + C group (*p* < 0.001). The effect size was small (*r* = 0.260) (Table 3). At follow-up, the proportion with increased exercise training was 52% in the PAP + C group and 30% in the PAP-only group.

### 3.3. Secondary Outcomes

No significant differences between the groups were shown for integration of physical activity as a habit at baseline (*p* = 0.265), follow-up (*p* = 0.738), or in the difference scores (*p* = 0.053) (Table 2).

Regarding HRQoL, no significant differences were shown in none of the eight health dimensions between the two groups at baseline. At follow-up, the PAP + C reported significantly higher levels in the two dimensions “physical functioning” (*p* = 0.016) and “physical role functioning” (*p* = 0.046) compared with the PAP-only group. The comparison of the difference scores showed that the improvement was significantly higher in PAP + C in four out of eight HRQoL dimensions; physical role functioning (*p* = 0.026), general health (*p* = 0.006), vitality (*p* = 0.020) and social role functioning (*p* = 0.022). The effect size for the differences were small and ranged between 0.118 and 0.144 (Table 2).

## 4. Discussion

This study has investigated differences in physical activity levels and HRQoL one year after PAP in patients who received physical counsellor support in addition to PAP and patients who did not accept this offer of support and only received PAP. The main finding was that patients who used physical activity counsellor support attained a higher level of weekly physical activity one year after receiving PAP than patients who did not use counsellor support. The increase in weekly physical activity among counsellor users was mainly due to increased exercise training rather than everyday activity.

These findings suggest that physical activity counsellors can influence patients to increase and maintain their physical activity. All the physical activity counsellors in the study were trained in motivational interviewing, an approach that can foster a more self-determined behavior and help patients overcome barriers to physical activity [27]. Motivational interviewing appears to support fulfilment of three basic needs described in Cognitive Evaluation Theory: competence, autonomy and social relatedness [28]. In our study, the counsellors accounted for the patients’ interests and priorities regarding physical activity, which supported competence to achieve the behavior change. The counsellors supported patients’ autonomy by exploring different behavioral options (e.g., different exercise opportunities) together with the patient. They also showed a genuine interest and expressed empathy, thus supporting social relatedness [29].

The model used for counselling support in addition to PAP in this study could be seen as a person-centered approach (also described as a client-centered approach) [30]. In a person-centered counselling approach, the PAP recipients become partners in planning, developing and monitoring care, and the counsellors support them to make changes and goals that best suit their personal needs and context [31,32]. Patient-centered counselling is a key component of the Swedish PAP program [11]. However, even though there are several similarities between a patient-centered approach and a person-centered approach, a person-centered approach broadens the perspective beyond the clinical perspective [31,32]. This perspective could be an important issue when it comes to supporting change in physical activity in every-day life [33].

Nearly two-fifths of the participants in this study took up the offer of counsellor support. Some patients may feel reluctant to seek support [34] and, therefore, it is likely that a PAP model in which patients are contacted by a counsellor leads to a higher proportion of counsellor use. For some, the reason for not utilizing the support of a physical activity counsellor might be related to factors that could be facilitated by the counsellors, such as low motivation to start physical activity [35] or insufficient self-efficacy and low expectations concerning one’s own ability to engage in physical activity [36]. Having more modest expectations of one’s own ability at the outset has been found to be related to achieving expected changes [37]. In this study there was no significant difference between the study groups in their confidence to change physical activity at baseline. However, the findings in this study among the non-counsellor users indicate that some patients can manage to increase their physical activity without support from physical activity counsellors. Nearly two-fifths among the non-counsellor users had increased physical activity one year after PAP.

The findings in this study showed that more participants among counsellor users reached the public health recommendation for physical activity for adults of at least 150 min of moderate to vigorous physical activity per week one year after PAP [38]. Among the counsellor users, 55% reached this physical activity level; this is less than the 66% who reach this level of activity in the Swedish adult population aged 18 to 64 years but similar to the level of activity in the population 65 years or older (54%) [39]. A previous study on patients receiving PAP within the studied health care organization showed that patients receiving PAP were older and had higher frequency of diagnoses than the general health care population [19]. The frequency of diagnoses among the participants in the present study was even higher and they were older. The treatment goal of physical activity for some patients, e.g., with physical limitations related to disease and for patients in older age groups, could be a small increase in physical activity and low intensity activity [40,41], whereas for other more healthy adults, a higher increase and high-intensity activity could be a possible goal. Other studies show that PAP is often used to support physical activity among older patients [13,42,43]. A small reduction in sedentary time [44] and low intensity activities [41] can have health benefits.

Another finding in this study was that the use of physical activity counsellor support in addition to PAP had positive effects on HRQoL. In this study, significant improvements were shown for four of eight health domains. Other studies of PAP interventions have reported significant positive effects of PAP on HRQoL in more or less [3,45,46] than the four health domains found in this study. The variation between PAP studies on effects on HRQoL outcomes may be a result of differences in study population characteristics with regard to patient morbidity [47], sex [48] and level of physical activity [5]. Furthermore, the specific PAP intervention investigated and the control group used, usual care [3,45] or written information about physical activity [46], may also influence differences in effects on HRQoL between PAP studies. The positive effects on HRQoL among counsellor users in this study could be a result of a higher increase in physical activity in this group [4,46], or as a result of improved positive health outcomes [40,46].

### 4.1. Limitations

The present study has some shortcomings that should be considered when interpreting the findings. The response rate at baseline was 40%. The non-responders at baseline were younger and more frequently prescribed specialized care (somatic or psychiatric). Of the baseline participants, 66% completed the follow-up. The dropouts at follow-up were younger, more frequently diagnosed with mental health disorders and less frequently diagnosed with circulatory diseases. However, our response rate of 66% at follow-up was higher compared to some previous PAP studies (62% vs. 56%) [4,49] but lower compared to others (83% vs. 73%) [5,48], including 6 respectively 12 months follow-up. Collecting data through postal questionnaires often yields low response rates. Although unwillingness to perform physical activity could be a reason for non-participation, other reasons might also apply, such as lack of interest or simply forgetting to respond [50]. Another limitation of the study was that we were not able to measure the baseline level of physical activity or HRQoL on the day when PAP was prescribed. Filling in the baseline questionnaire a few weeks after prescription might influence self-reporting on baseline physical activity. However, the primary aim was to identify differences in physical activity and HRQoL one year after PAP between the two groups (counsellor use or not), and there were no differences in how baseline and follow-up data were collected for both groups. Still, the effect size might have been higher if the reported baseline activity level was overestimated. The overall results of this study must be interpreted with caution regarding generalization to other clinical populations and health care contexts. Characteristics of PAP recipients, clinical populations and prescribing health care contexts influence the generalizability of study results.

These limitations should be balanced against the study’s strengths. The study concerned a real-world PAP intervention, designed to include all patients receiving PAP over a period of one year in primary care and specialized somatic and psychiatric routine health care. This type of real-world intervention is important for developing the evidence base for PAP provided in routine clinical practice, i.e., evidence based on effectiveness rather than efficacy studies. However, patient-related selection bias regarding willingness to participate is difficult to overcome in this type of study. Regardless, the study included a relatively large sample of a clinical PAP population. The fact that the recruitment process lasted for one year reduced the risk of bias associated with seasonal variation in weather, which is known to influence maintenance of physical activity [51]. Recruitment of participants and data collection were handled by one of the researchers, which meant that health care professionals and counsellors were not interrupted by study-related tasks. Hence, PAP could be provided to patients as part of a real-world routine health care delivery of PAP.

### 4.2. Implications and Future Research

To our knowledge, this is the first study in Swedish routine care investigating a PAP model in which counsellor support was offered in addition to health care professionals’ delivery of PAP. Previous studies involving the Swedish PAP program have described models using counsellor support [5,48], but no study has evaluated the potentially added benefits of using counsellor support versus not using this support. The result concerning physical activity in the PAP-only group in this study indicates that some patients do not need to take part of the offered counsellor support. Studies of PAP commonly investigate the effectiveness of the program as a whole and patients who do not complete the program after PAP are, despite an increase in physical activity, considered non-adherers [13]. In further evaluations of PAP programs, we suggest that greater focus should be placed on investigating what different program components add to patients’ physical activity after PAP. PAP programs are often adapted to fit local health care contexts, underscoring the importance of transparency to enable description and analysis of the contents of the programs. Regarding counsellor support in addition to PAP, further studies are needed to provide evidence for the effectiveness of different patient support models in addition to health care professionals PAP. It is also important to identify which patients would benefit from counsellor support.

## 5. Conclusions

Patients who used physical activity counsellor support in addition to receiving PAP from health care professionals had higher physical activity and better HRQoL one year after prescription of physical activity compared with patients who did not use this support. The increase in physical activity among counsellor users was mainly due to increased exercise training rather than everyday activity.

## Figures and Tables

**Figure 1 ijerph-17-05679-f001:**
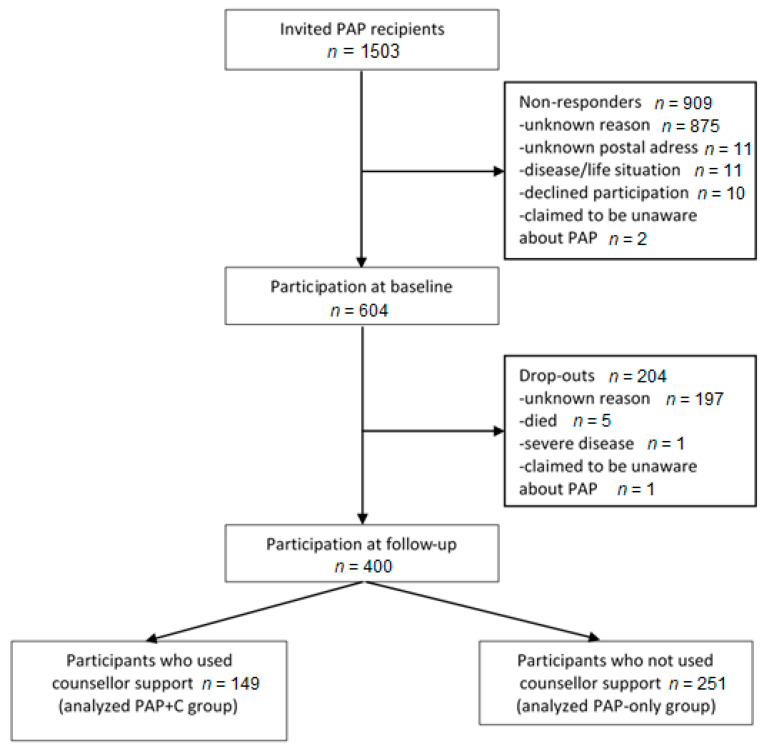
Flowchart of the recruitment process.

**Table 1 ijerph-17-05679-t001:** Characteristics in all participants at baseline and differences in characteristics between the study groups.

	All	PAP + C	PAP-only	
Characteristics	*n* = 400	*n* = 149	*n* = 251	*p*-Value
Age, mean (SD) (min-max)	62.0 (14.0) (18–90)	59.6 (15.2) (18–90)	63.5 (13.1) (21–90)	0.008 ^a^
Female sex, *n* (%)	276 (69.0)	99 (66.6)	177 (70.5)	0.394 ^b^
University education, *n* (%)	103 (26.7)	48 (33.6)	55 (22.6)	0.019 ^b^
Confidence to change (scale 1–10), median (IQR)	7.0 (3.0)	7.0 (3.0)	7.0 (3.0)	0.211^c^
Important to change (scale 1–10), median (IQR)	8.0 (4.0)	8.0 (3.0)	8.0 (4.0)	<0.001^c^
Common occurring diagnosis, *n* (%) ^c^				
Musculoskeletal diseases	239 (61.6)	80 (54.8)	159 (64.9)	0.047 ^b^
Circulatory diseases	207 (52.9)	80 (54.8)	127 (51.8)	0.571 ^b^
Endocrine diseases	156 (39.9)	55 (37.7)	101 (41.2)	0.488 ^b^
Mental health disorders	101 (25.8)	36 (24.7)	65 (26.5)	0.682 ^b^
Respiratory diseases	104 (26.6)	44 (30.1)	60 (24.5)	0.222 ^b^
Number of diagnostic groups, *n* (%) ^d^				0.057 ^b^
1–4	105 (27.1)	47 (32.6)	58 (23.8)	
≥5	283 (72.9)	97 (67.4)	186 (76.2)	
Frequency of health care visits, *n* (%) ^e^				0.137 ^b^
1–10	160 (40.8)	67 (45.6)	93 (38.0)	
≥11	232 (59.2)	80 (54.4)	152 (62.0)	
≥1 overnight stays in hospital, *n* (%) ^e^	99 (25.2)	41 (28.0)	58 (24.0)	0.340 ^b^
Prescribing professional, *n* (%)				0.046 ^b^
Physician	111 (27.8)	50 (33.6)	61 (24.3)	
Other e.g., nurse, physiotherapist	289 (72.3)	99 (66.4)	190 (75.7)	
Prescribing setting, *n* (%)				0.646 ^b^
Primary care	332 (83.0)	122 (81.9)	210 (83.7)	
Specialized somatic or psychiatric care	68 (17.0)	27 (18.1)	41 (16.3)	

PAP + C; patients receiving physical activity prescription + counsellor support. PAP-only; patients receiving physical activity prescription without counsellor support. ^a^ Independent sample *t*-test; ^b^ Pearson Chi-square test; ^c^ Mann Whitney U test; ^d^ Medical record registrations 12 months prior PAP of the International Classification of Diseases version 10 (ICD 10) classified diagnostic groups excluded O (pregnancy, childbirth) and Z (factors influencing health status/health care contacts); ^e^ Medical record registrations 12 months prior PAP of health care visits in primary and/or specialized somatic and psychiatric care with no selection of professional; overnight hospital stays in somatic or psychiatric care.

**Table 2 ijerph-17-05679-t002:** Differences between the study groups in weekly physical activity, physical activity habit and health-related quality of life (HRQoL) at baseline and follow-up, and differences in effects one year after PAP.

	Baseline			Follow-up			Difference between Baseline and Follow-up				
Outcomes	Median (IQR)	Meanrank	*p*-Value ^a^	Median (IQR)	Meanrank	*p*-Value ^a^	Median Diff. (IQR)	Meanrank Diff.	*p*-Value ^b^	Effect Size ^c^	Valid Data
Weekly physical activity (3–19)			0.001			0.009			<0.001	0.259	
PAP + C	9.0 (4.0)	154.3		11.0 (7.0)	196.1		2.0 (7.0)	211.8			*n* = 135
PAP-only	11.0 (5.0)	192.6		9.5 (6.0)	166.9		0.0 (4.0)	157.3			*n* = 220
Physical activity habit (1–7)			0.265			0.738			0.053	0.120	
PAP + C	3.3 (2.8)	124.4		4.0 (2.3)	132.3		0.5 (2.1)	141.1			*n* = 109
PAP-only	4.0 (3.0)	134.9		4.0 (2.5)	129.2		0.0 (1.5)	112.9			*n* = 151
Health Related Quality of Life (0–100)											
Physical functioning											
PAP + C	75.0 (33.8)	195.7	0.136	80.0 (30.0)	202.3	0.016	0.0 (6.0)	189.5	0.529	0.033	*n* = 138
PAP-only	70.0 (35.0)	178.6		70.0 (38.9)	174.7		0.0 (6.0)	182.3			*n* = 231
Physical role functioning			0.773			0.046			0.026	0.119	
PAP + C	50.0 (100)	172.1		75.0 (75.0)	187.1		0.0 (25.0)	188.5			*n* = 132
PAP-only	50.0 (100)	175.2		50.0 (100)	166.0		0.0 (25.0)	165.1			*n* = 215
Bodily pain			0.264			0.212			0.789	0.014	
PAP + C	56.3 (49.4)	194.0		62.5 (45.0)	195.0		0.0 (24.3)	184.1			*n* = 138
PAP-only	45.0 (45.0)	181.2		55.0 (52.5)	180.7		0.0 (26.3)	187.1			*n* = 233
General Health			0.701			0.137			0.006	0.144	
PAP + C	60.0 (30.0)	176.8		65.0 (35.0)	190.1		5.0 (25.0)	198.6			*n* = 135
PAP-only	55.0 (30.0)	181.1		55.0 (35.0)	173.2		0.0 (20.0)	168.0			*n* = 223
Vitality			0.253			0.408			0.020	0.122	
PAP + C	55.0 (32.5)	174.9		65.0 (30.0)	188.9		5.0 (20.0)	199.6			*n* = 137
PAP-only	60.0 (35.0)	187.9		60.0 (35.0)	179.5		0.0 (25.0)	173.1			*n* = 228
Social role functioning			0.163			0.726			0.022	0.118	
PAP + C	75.0 (50.0)	177.1		75.0 (37.5)	189.5		0.0 (25.0)	203.1			*n* = 138
PAP-only	75.0 (37.5)	192.8		75.0 (37.5)	185.6		0.0 (25.0)	177.5			*n* = 235
Emotional role functioning			0.711			0.714			0.415	0.044	
PAP + C	100 (66.7)	172.2		100 (66.7)	176.8		0.0 (33.3)	179.6			*n* = 133
PAP-only	100 (66.7)	176.0		100 (66.7)	173.1		0.0 (0.0)	171.3			*n* = 215
Mental Health			0.396			0.870			0.170	0.072	
PAP + C	72.0 (32.0)	175.1		76.0 (28.0)	182.2		0.0 (18.0)	190.6			*n* = 137
PAP-only	76.0 (32.0)	184.6		76.0 (35.5)	180.3		0.0 (16.8)	175.1			*n* = 224

PAP + C; patients receiving physical activity prescription + counsellor support. PAP-only; patients receiving physical activity prescription without counsellor support. ^a^ Mann-Whitney U test; ^b^ Average difference score between the baseline- and follow-up assessment; ^c^ Interpretation of Cohen (1988) *r* = Z/√2; small effect 0.10, medium effect 0.30, large effect 0.50.

**Table 3 ijerph-17-05679-t003:** Differences between the study groups in everyday activity and exercise training at baseline and follow-up, and differences in effects one year after PAP.

Everyday Activity and Exercise Training	Baseline			Follow-up			Difference in Effects between Baseline and Follow-up			
	PAP + C	PAP-only		PAP + C	PAP-only		PAP + C	PAP-only		
Everyday activity	***n*** **= 147**	***n*** **= 249**	***p*** **-Value ^a^**	***n*** **= 143**	***n*** **= 242**	***p*** **-Value ^a^**	***n*** **= 141**	***n*** **= 241**	***p*** **-Value ^a^**	**Effect Size ^c^**
Mean rank	192.4	202.1	0.404	209.4	183.3	0.024	209.2	181.1	0.014	0.126
Categorical response, *n* (%)										
1. 0 min	4 (2.7)	6 (2.4)		1 (0.7)	2 (0.8)					
2. <30 min	15 (10.2)	20 (8.0)		9 (6.3)	21 (8.7)					
3. 30–60 min	25 (17.0)	36 (14.5)		19 (13.3)	36 (14.9)					
4. 60–90 min	17 (11.6)	35 (14.1)		17 (11.9)	45 (18.6)					
5. 90–150 min	29 (19.7) ^b^	53 (21.3) ^b^		30 (21.0) ^b^	54 (22.3) ^b^					
6. 150–300 min	34 (23.1)	50 (20.1)		39 (27.3)	50 (20.7)					
7. >300 min	23 (15.6)	49 (19.7)		28 (19.6)	34 (14.0)					
Exercise training	*n* = 146	*n* = 238	*p*-value ^a^	*n* = 139	*n* = 234	*p*-value ^a^	*n* = 137	*n* = 222	*p*-value ^a^	Effect size ^c^
Mean rank	167.8	207.6	<0.001	203.3	177.3	0.021	213.45	159.36	<0.001	0.260
Categorical response, *n* (%)										
1. 0 min	63 (43.2)	67 (28.2)		31 (22.3)	72 (30.8)					
2. <30 min	32 (21.9) ^b^	40 (16.8)		23 (16.5)	44 (18.8)					
3. 30–60 min	26 (17.8)	64 (26.9) ^b^		33 (23.7) ^b^	50 (21.4) ^b^					
4. 60–90 min	10 (6.8)	36 (15.1)		16 (11.5)	32 (13.7)					
5. 90–120 min	7 (4.8)	19 (8.0)		16 (11.5)	11 (4.7)					
6. >120 min	8 (5.5)	12 (5.0)		20 (14.4)	25 (10.7)					

PAP + C; patients receiving physical activity prescription + counsellor support. PAP-only; patients receiving physical activity prescription without counsellor support. ^a^ Independent Mann–Whitney U test; ^b^ Median category for each group; ^c^ Interpretation of Cohen (1988) *r* = Z/√2; small effect 0.10, medium effect 0.30, large effect 0.50.

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
