# Peer review of "Physical Activity on Prescription in Routine Health Care: 1-Year Follow-Up of Patients with and without Counsellor Support"

_ijerph, 2020, doi:10.3390/ijerph17165679_

Round 1

Reviewer 1 Report

“Physical activity on prescription in routine health care: 1-year follow-up of patients with and without counsellor support.”

The study aims to identify the influence of counsellor support on physical activity levels and health-related quality of life one year after starting with physical activity on prescription (PAP). For this purpose, the study compare patients who assumed this support with patients who have not and focus on a 1-year follow-up. This comparison clearly demonstrates that additional support has significant impact on physical activity level and “partly” on health-related quality of life one year after starting PAP.

In general, the manuscript is well written, deals with an interesting topic and is comprehensibly structured. Nevertheless, there are some points that should be adjusted so that the reader can better understand the methodology/results. This refers on one hand to the chapter Material and Methods, which should be divided into subcategories, and on the other hand to the tables, which are partly difficult to read.

Page 1, line 28: I would suggest to briefly adress the results of the secondary endpoint (HRQoL) in one sentence in the abstract.

Page 2, line 60: For better readability, it would be useful to separate the chapter into sub-chapters (e.g. participants, outcome measures, statistics)

Page 2, line 66:200,000 people = total sum of people that have received healthcare?, 200,000 per year? The population in the community? Please specify!

And, what is meant with healthcare organization? Is this a company, or another kind of association? Please specify!

Which health care settings have been addressed here? Table 1 suggests that there are differences (specialized somatics…), please specify.

Page 2, line 69: Here, the compliance with the Declaration of Helsinki and the ethical approval (including number) should be mentioned.

Page 2, line 74: you state that different modes of delivery exist, which modes are these?

Page 2, line76: please specify “multiprofessional organization”, who is part of that?

page 2, line83: How have the healthcare professionals been trained in motivational interviewing? Have there been specific educational sessions? How many and how long, and does this follow a specific curriculum? Please add some information about the structure of this training!

Page 3, line 100:  I would suggest to report the mean values and standard deviation as 3.5 ± 1.8,  and then name the wide range from 1 to 12.

Page 3, Figure 1: According to the flow diagram of the CONSORT-Line you may add the type of statistical analysis below both groups (e.g. ITT-Analysis, PP-Analysis).

Page 4, line 141ff.: What was the purpose of the data collection 12 month before and 16 month after PAP, when the primary focus is a 1-year follow-up? Please clarify.

Page 4, line 152 and 154: you may put “1” and “6” in quotes (“1” implies no exercise while “6” implies more than….); also in the other lines.

Page 4, line 159: Delete “to” before 150 minutes.

Page 4, line 159 and 149: Do the mentioned questions allow a differentiation between moderate and high intensities in physical activity?

Page 5, line 197: After “The” a word is missing. I would place the following sentence further ahead. See previous comment.

Page 5, line 199 ff.: The written presentation of the results should be supplemented by Table 1, which is why it’s not necessary to repeat everything in continuous text. Therefore, I would suggest to limit the text to percentages and significances and show the absolute values only in the table.

Page 5, line 207: I couldn’t find the information about counselling visits in Table 1.

Page 5, Table 1:

What is the rationale in classifying in 1-4 and > 5 diagnostic groups? What is meant by diagnostic groups? Is this the number of diagnosed diseases and does this correspond to multimorbidity?

Ditto regarding health care visits, what is the rationale of this categorization? Are the p-values related to total numbers, and what is the purpose of the categorization?

What does the column “valid n” add to the table? May be deleted.

Page 6, line 217 ff: For better readability chapter 3.1 should be split and assigned to the previous or subordinate chapters. The description of baseline differences of age, educational level and prescribed PAP (line 218 -221) will fit with the previous part describing the study population and the significant differences in primary and secondary outcomes (line 222 – 227) should be placed in the respective chapters.

Page 7, line 230, table 2and 3: you report mean ranks with quite a high number. Are these really mean ranks or are these the rank sums that have been used to calculate U in the Mann-Whitney test? Please check this.

Page 7, Table 2: In Table 2, the median and mean rank are missing in the category Physical functioning for PAP+C in follow-up. Why, and how have the significance tests been calculated without these values?

Replace 9,5 in the category weekly physical activity/PAP-only/follow-up with 9.5.

Page 8, Table 3: The units in the tables are not entirely clear to me. Are we talking about minutes or other categories in case of weekly everyday activity and weekly exercise training? Please check this in all tables and add it accordingly. Also check the Categorical responses, as it’s sometimes more or less than 100% in total.

Page 8, Line 236: different font

Page 9, line 243: you might delete “keep close to a medium effect”, as this might appear as a desired result

Page 9, Line 244: add a ) after (ES=0.26)

Page 9, line 252ff: Are the described percentage increases and decreases in the respective groups significant? If so, please add the p-value.

Page 9, line 260: ES-value (ES=0.012) differs from ES-value in table 2 (ES=0.120).

Page 10, line 328: Another limitation is, that you haven’t assessed motivational-volitional differences between the two groups. What could have been the motivation of the participants in the PAP+C group to choose counselling? Also here might be a systematic difference between the two groups!

Page 10, line 333: How exactly do the follow-up response rate differ from other studies? A numerical comparison is useful here, so that one can better classify the 66%-rate in this study.

Page 10, line 359: Are there studies in other countries (e.g. New Zealand) that also perform a kind of PAP with voluntary counsellor support and have made such comparisons?

Page 10, line 370: What consequences do the findings have for PAP in Sweden or other international models? What can other countries learn from this study when developing such models?

Reviewer 2 Report

Abstract - what do the numbers correspond to? - activity score - need to explain

Overall a well written article with clear findings. Some points for clarification or enhancement are suggested below. 

Abstract

It's not clear until you read the full paper what the median scores mean - I think this relates to change in median activity score but perhaps needs to be explained.  

Background

Line 39/40 - can you make this point a bit clearer as I didn't really follow it. 

This section includes some very long sentences which could benefit from being broken into separate sentences to enhance ease of reading - e.g. lines 54-57.

I think a couple more lines are needed to describe the addition of counselling support and what that entails for patients. Is this offered as part of routine care across Sweden?  

Method

The description of the study as an "observational prospective follow-up" is a little confusing.  Whilst I understand what you mean, the terms 'prospective' and 'follow-up' are instinctively contradictory. Is there an easier way to explain this?

What constitutes a patient “in need” of physical activity prescription? Some additional context would be useful.  

The information on numbers of non-responders and dropouts - would this be better positioned within the results section?

Results

ES please write effect size in full when first used and then abbreviate henceforth

Line 263: I'd suggest removal of the words “more importantly” since that could be considered subjective

Discussion

Overall, the discussion section could benefit from more development to situate the findings within the wider body of literature on PAP.  

A section summarising the implications of the findings would enhance this discussion. For example, what do these findings mean for healthcare services and systems beyond Sweden? How might these findings be relevant to international readers? A little more detail on how counselling support is currently funded as part of routine care would be useful - if the findings are to influence other healthcare services to consider implementing PAP+C. 

Lines 277-283.  I would recommend more tentative language when discussing the mechanisms by which MI might contribute to study findings. As far as I can tell there was no treatment fidelity component or use of MI coding framework to see whether MI skills were consistently or effectively applied?

Lines 303-316.  Can you make this point clearer? I was unsure how it relates to the study findings. 

Lines 299-302 acknowledge 2/5ths without counsellor support showed increase PA at 1 year - could you go further in your discussion or link to conclusions to suggest why healthcare services should therefore invest in offering counselling support?

Lines 325-327 - is it also possible that HRQoL improved not as a result of increase in PA but because of wider benefits associated with regular health counselling - e.g. feeling valued, increased interest in health and lifestyle factors?

The changes in HRQoL are interesting - this section of the discussion needs more development.
